# The Influence of Met Receptor Level on HGF-Induced Glycolytic Reprogramming in Head and Neck Squamous Cell Carcinoma

**DOI:** 10.3390/ijms21020471

**Published:** 2020-01-11

**Authors:** Verena Boschert, Nicola Klenk, Alexander Abt, Sudha Janaki Raman, Markus Fischer, Roman C. Brands, Axel Seher, Christian Linz, Urs D. A. Müller-Richter, Thorsten Bischler, Stefan Hartmann

**Affiliations:** 1Department of Oral and Maxillofacial Plastic Surgery, University Hospital Würzburg, D-97070 Würzburg, Germany; 2Department of Biochemistry and Molecular Biology, Theodor Boveri Institute, Biocenter, University of Würzburg, D-97074 Würzburg, Germany; 3Core Unit Systems Medicine, University of Würzburg, D-97080 Würzburg, Germany; 4Interdisciplinary Center for Clinical Research, University Hospital Würzburg, D-97080 Würzburg, Germany

**Keywords:** HNSCC, head and neck cancer, HGF, Met, cancer metabolism

## Abstract

Head and neck squamous cell carcinoma (HNSCC) is known to overexpress a variety of receptor tyrosine kinases, such as the HGF receptor Met. Like other malignancies, HNSCC involves a mutual interaction between the tumor cells and surrounding tissues and cells. We hypothesized that activation of HGF/Met signaling in HNSCC influences glucose metabolism and therefore substantially changes the tumor microenvironment. To determine the effect of HGF, we submitted three established HNSCC cell lines to mRNA sequencing. Dynamic changes in glucose metabolism were measured in real time by an extracellular flux analyzer. As expected, the cell lines exhibited different levels of Met and responded differently to HGF stimulation. As confirmed by mRNA sequencing, the level of Met expression was associated with the number of upregulated HGF-dependent genes. Overall, Met stimulation by HGF leads to increased glycolysis, presumably mediated by higher expression of three key enzymes of glycolysis. These effects appear to be stronger in Met^high^-expressing HNSCC cells. Collectively, our data support the hypothesized role of HGF/Met signaling in metabolic reprogramming of HNSCC.

## 1. Introduction

Head and neck squamous cell carcinoma (HNSCC) is a highly challenging malignancy accounting for more than 650,000 new cases of cancer worldwide each year [1]. There are several known risk factors, such as exposure to tobacco and alcohol, as well as poor oral hygiene and infection with high-risk human papilloma viruses (HPV). Comprehensive survival rates for HNSCCs are becoming less valid, since the tumor site, HPV status, and treatment regime have a high impact on survival. There is evidence for distinct variation in the genetic landscape of HPV-positive and HPV-negative HNSCCs, reflecting reasons for different treatment responses and outcomes of patients [2]. However, the hepatocyte growth factor (HGF) and its sole receptor, Met, seem to be important for both HPV-positive and -negative tumors. Seiwert and colleagues showed Met overexpression in approximately 90% of human tumor tissue samples and HNSCC cell lines [3]. Increased gene copy numbers or genetic mutations of Met are rather rare in HNSCC, so other mechanisms like transcriptional modulation could often be the cause for receptor overexpression [4]. Additionally, the ligand HGF was found to be overexpressed in almost 50% of human tumor tissue samples. HGF is not secreted by the tumor cells itself, but by tumor-associated fibroblasts [5]. Collectively, alteration and aberrant activation of HGF/Met signaling in HNSCC is correlated with poor prognosis and shorter overall survival in HNSCC patients [4,6,7].

The HGF/Met pathway is known to contribute substantially to proliferation, migration, invasion, metastasis, and anti-apoptotic signaling in HNSCC [8]. Furthermore, HGF plays a role in metabolic reprogramming of tumor cells. HGF enhanced expression of glucose transporters GLUT-1 and GLUT-4 in myocytes, increased glucose consumption and lactate production in a breast cancer model, and induced hexokinase 2 (HK2) expression in a lung cancer model [9,10,11]. These changes help to maintain aerobic glycolysis, also known as the Warburg effect, in cancer cells [12]. High glucose consumption and metabolism in HNSCCs is clinically relevant, since the uptake of (18)F-2-fluoro-2-deoxy-d-glucose (FDG), as used in PET/CT scans, is significantly associated with the expression of key molecules of glucose metabolism (GLUT, HK 2, HIF1a) [13].

Extensive glucose metabolism is thought to be beneficial for tumor cells [14,15]. Even if the yield of ATP is much higher when the cells perform oxidative phosphorylation (OXPHOS), adherence to aerobic glycolysis has some major advantages for the tumor cell. Glycolysis is clearly faster than OXPHOS, it generates more biosynthesis-relevant precursor molecules [14,15], and it has an impact on the tumor microenvironment, such as fibroblasts and cytotoxic T cells. Notably, high levels of lactate, which is the end product of aerobic glycolysis, suppress the activation and proliferation of cytotoxic T cells [16].

Importantly, Met knockdown led to impaired lymph node metastasis and prolonged survival in an in vivo HNSCC model [17]. Furthermore, primary tumors of patients with advanced lymph node spread (N2/N3) showed higher Met levels than those with early stage disease (N0/N1) [18]. Met expression was typically found to be higher in metastases compared to the corresponding primary tumors [18]. This illustrates the importance of HGF/Met signaling for metastatic spread and therefore underscores the potential of Met-targeting drugs in an advanced or metastatic stage disease of HNSCC.

The aim of our study was to evaluate the impact of HGF/Met signaling on glucose metabolism in HNSCC. Additionally, we addressed the question whether the Met expression level is relevant to the amount of metabolic reprogramming. This is important because targeted therapies, e.g., Met inhibition by tyrosine kinase inhibitors, are used in patients with advanced stage or recurrent/metastatic disease.

## 2. Results

To determine the impact of HGF/Met signaling on glycolysis in head-and-neck tumor cells in vitro, we used three established cell lines, one from a primary hypopharyngeal site (FaDu), one from a metastatic site originating from a tumor of the pharynx (Detroit 562), and a third from a primary tumor of the tongue (SCC-9). To check whether the Met receptor is activated when HGF (50 ng/mL) was added to the cells, we performed an immunoblot to detect activated Met receptor via an antibody specific for Met phosphorylated at residues Tyr1234/Tyr1235 (Figure 1). In all three cell lines, a strong signal only occurred with this antibody when cells had been stimulated with HGF. The addition of tyrosine kinase inhibitor PHA-665752 abrogated the signal, showing the HGF-specificity of the results. Next, we wanted to check whether HGF/Met signaling had an impact on the growth of the cell lines. Therefore, we subjected them to two different tyrosine kinase inhibitors known to inhibit Met. Both PHA-665752 and foretinib led, concentration-dependently, to a weaker signal in a crystal violet assay (Figure 2). In all three cell lines, foretinib exhibited an IC_50_ value of 2–3 µM, whereas PHA-665752 had a lower efficacy, with an IC_50_ of approximately 8 µM. Furthermore, stimulating the cell lines with HGF had a small, but significant, proliferating effect on Detroit 562 and SCC-9 cells, but only a nonsignificant effect on FaDu cells, as shown in the results of a crystal violet-based proliferation assay (Figure 3).

To get an idea which genes were regulated in these cell lines upon HGF stimulation, we performed mRNA sequencing (Appendix A). Therefore, cells were stimulated with HGF for 16 h or were left untreated. A Venn diagram of genes that were substantially up- or downregulated upon HGF stimulation in comparison to untreated controls shows that the cell lines reacted quite differently (Figure 4a). In Detroit 562, a total of 2850 genes were regulated, whereas in FaDu only 236 and in SCC-9 only 30 genes showed significantly different expression levels in comparison to the control (adjusted *p*-value *p*_adj_ < 0.1). Furthermore, FaDu and Detroit 562 shared 171 regulated genes and Detroit 562 and SCC-9 cells 21. The four differentially expressed genes (Glycine transporter SLC6A9, GTP binding protein GBP1, Jun dimerization protein JDP2, and TNFα-induced protein TNFAIP2) shared by FaDu and SCC-9 cells were also regulated in Detroit 562. We examined the expression levels of the Met receptor and found that the Met mRNA concentration was significantly higher in Detroit 562 than in the other two cell lines (Figure 4b). This is comparable to Western blot results (Figure 4c), which also showed a higher concentration on the protein level in this cell line. Stimulation of cells with HGF did not lead to significant differences in the expression of the gene (Figure 4b).

To get an idea which pathways in the cells are regulated the most upon HGF stimulation, we performed functional enrichment analysis based on genes with significant differences in expression levels in our mRNA sequencing analysis (see Materials and Methods). Table 1 shows a list of the most upregulated signaling pathways in Detroit 562. Among these are several ones linked to cell proliferation, e.g., Ribosome biogenesis. Furthermore, genes for laminins and integrins, which belong to the focal adhesion gene group are highly upregulated, contributing to a higher cell motility. The pathways for MAPK and PI3K-Akt are also in the top ten pathways reported to be activated upon MET activation and to contribute to proliferation and differentiation. Altogether the most upregulated pathways are in accordance with the reported effects of Met receptor activation. Functional classes enriched in downregulated genes include several pathways for degradation of different substances, like xenobiotics, chemical carcinogens, porphyrine, and chlorophyll (Table 2). These pathways have in common different glutathione S-transferases, glucuronyl- and glucuronosyltransferases, alcohol dehydrogenases, and aldehyde dehydrogenases, which are significantly downregulated upon HGF stimulation. For a more detailed look at the results of the enrichment analysis the complete tables can be found in the Appendix A. Of the 287 pathways with at least one significantly upregulated gene in Detroit 562, 44 pathways are significantly upregulated (*p*.adjust < 0.1, Appendix A) and of the 311 with at least one significantly downregulated gene 29 pathways are significantly downregulated (*p*.adjust < 0.1, Appendix A). In contrast to Detroit 562, less pathways are significantly regulated in the other two cell lines (FaDu 22 upregulated, none downregulated, SCC-9 6 upregulated, 7 downregulated, see Appendix A).

As HGF is known to play a role in metabolic reprogramming, we were interested in the mRNA sequencing results for the KEGG pathway hsa00010 (glycolysis/gluconeogenesis). Four genes were upregulated, and enrichment analysis resulted in an adjusted *p*-value of 0.71 for this pathway (Table 1). Ten genes were downregulated and accounted for an adjusted *p*-value of 0.41 (Table 2). Figure 5a shows a heat map of expression changes of genes associated with hsa00010. Genes quoted to be regulated in Table 1 and Table 2 are marked with asterisks. One of the upregulated genes is HK2, which codes for hexokinase II, an enzyme catalyzing the first, rate-limiting step of glycolysis, where glucose is phosphorylated to glucose-6-P. Detroit 562 showed the lowest HK2 expression in the control sample out of the three cell lines (Figure 6a). Stimulation with HGF led to an increase (log_2_ fold change of 0.64, *p*_adj_ < 0.1) of the mRNA concentration of this gene, leading to expression levels comparable to those of the other two cell lines.

HGF stimulation had also weak but no significant effects on HK2 expression in FaDu and SCC-9, with log_2_ fold changes of 0.398 and 0.129, respectively. Western blot of cell lysates after 48 h of stimulation with HGF confirmed the different expression levels in the three cell lines. Expression was comparable to control when tyrosine kinase inhibitor Foretinib was added (Figure 6b). Another important enzyme participating in glycolysis is lactate dehydrogenase (LDH), which catalyzes the conversion of pyruvate to lactate or vice versa. The gene for subunit LDHA was upregulated in Detroit 562 upon HGF stimulation (Figure 5a and Figure 6c, log_2_ fold change of 0.43, *p*_adj_ < 0.1). Also on protein level an increase of LDHA was observable (Western Blot, Figure 6d), but only with HGF alone, not in the presence of Foretinib.

The genes that were downregulated the most coded for alcohol dehydrogenases. ADH7 was downregulated (log_2_ fold change −3.75, *p*_adj_ < 0.1) in Detroit 562, and its expression in the absence of any stimulation was low in all three cell lines (Figure 5a, and Appendix A). Genes ADH1B and ADH1C were highly expressed in Detroit 562 cells, and stimulation with HGF led to a strong decrease in expression (Figure 5 and Figure 6e and Appendix A, log_2_ fold changes of −3.16 and −3.14, respectively). For ADH1B a Western blot confirmed that protein concentration was also reduced upon HGF stimulation (Figure 6f). Furthermore, no decrease in protein concentration was observed in the presence of 500 nM Foretinib.

To examine several other genes contributing to carbon metabolism, especially in cancer, we generated a heat map using KEGG pathway hsa05230 (Central carbon metabolism in cancer, Figure 5b). Nine genes were upregulated and enrichment analysis resulted in an adjusted *p*-value of 0.07 (Table 1), showing that this pathway is significantly upregulated (*p*.adjust < 0.1). Nine genes were downregulated and accounted for a total adjusted *p*-value of 0.58 (Table 2). The strongest effect on gene expression was found on two genes of the FGFR family. The gene of FGFR1 was significantly upregulated in Detroit 562 (log_2_ fold change of 1.32), whereas the gene of FGR3 was significantly downregulated (log_2_ fold change of −1.11, Figure 5b and Figure 7a,c). For both receptors we were able to confirm these changes using quantitative PCR (Figure 7b,d). We were not able to investigate the receptors using Western blot in our cells, probably due to concentrations not high enough to be detected by the antibody.

To investigate if all these expression changes of glycolysis and cancer-relevant genes upon HGF stimulation led to changes in glucose metabolism in Detroit 562, we performed extracellular flux assays (glycolytic rate assays). Proton efflux and oxygen consumption can be measured in the supernatant of cells in several samples simultaneously and in real time with this kind of assay. The measured oxygen consumed accounts for the OXPHOS that has taken place in the cells and, in combination with an inhibition of mitochondrial function by rotenone and antimycin A, is used for calculating the fraction of proton efflux derived by CO_2_ from OXPHOS (oxidative phosphorylation). Subtraction of this mitochondrial acidification from total proton efflux results in the glycolytic proton efflux rate (glycoPER). All three cell lines were measured after 48 h HGF stimulation, HGF stimulation plus PHA-665752 treatment, or control treatment with medium alone. In Detroit 562, a notable effect of HGF on glycoPER was seen in comparison to the control, resulting in a higher value for basal and compensatory glycolysis compared to the control (Figure 8a). PHA-665752 treatment prevented this in part. FaDu cells also showed this increased glycolysis, but only to a minor extent, whereas SCC-9 cells show no increase at all (Figure 8a,c). These results demonstrate that HGF can indeed influence the glycolytic rate of head and neck cancer cells but that not every cell line responds to the same extent.

## 3. Discussion

It has been widely shown in several in vitro and in vivo studies that HGF/Met signaling contributes to malignant properties in HNSCC [8]. Clinical data underscore that overexpression of HGF and/or Met results in shorter overall survival of HNSCC patients [4,6,7]. In contrast, in a phase 1/2 trial with foretinib, a Met-targeting tyrosine kinase inhibitor, the best response was only stable disease, and phase 2 was not entered (NCT00725764). However, HGF/Met signaling is still considered clinically relevant, and there is an ongoing investigation of ficlatuzumab (a HGF-targeting monoclonal antibody) in a recruiting phase 2 trial in cetuximab-resistant R/M HNSCC patients (NCT03422536).

For the present study, we aimed to investigate the role of HGF/Met signaling in HNSCC cells with different levels of Met receptor expression. Since Met is a crucial driver of metastasis, we expected the highest Met expression level in a cell line derived from metastatic spread. We confirmed this hypothesis by Western blot analysis, showing the strongest signal for total Met in Detroit 562 cells (Figure 4c). Importantly, this cell line was generated from a distant metastatic site in a HNSCC patient (pleural effusion, [19]). Further investigation showed a Met gene copy number amplification in Detroit 562, further suggesting a clonal selection of Met^high^ tumor cells during the course of metastatic spread (data derived from cbioportal.com, [20]). Copy number changes are rare events in primary HNSCC tumors. However, systemic therapies, such as chemotherapy or targeted therapy, typically target locoregional or distant metastases, which show higher Met levels than the primary tumor.

Importantly, as shown in Figure 2 and Figure 3, the growth-stimulating effects of HGF and the efficacy of Met-directed tyrosine kinase inhibitors, such as PHA-665752 and foretinib, seem to be independent of Met expression levels. The IC_50_ values for foretinib were 3.1 µM in SCC9, 2.3 µM in Detroit 562, and 2.8 µM in FaDu. In another HNSCC model, comparable IC_50_ numbers were reported for crizotinib, a Met-directed tyrosine kinase inhibitor [5]. In their set of cell lines, a cell line derived from a locoregional metastatic site (UM-SCC22B) was included, and no differences in IC_50_ values were observed. Taken together, these findings suggest that changes in Met copy numbers can be present but do not necessarily influence the IC_50_ values of Met-directed tyrosine kinase inhibitors.

The effects of HGF on proliferation were quite low for all three cell lines (Figure 3). The increases in proliferation also do not correspond to the observed Met receptor levels, probably because a lot of other growth influencing pathways like, for example, other tyrosine kinases exist in different levels in the cells. These pathways could be already active, especially when grown in full FCS (fetal calf serum)-containing media, leading only to small increases when triggered by additional factors. However, the effects of HGF stimulation on gene upregulation determined by unbiased mRNA sequencing showed dramatic differences between the three cell lines used in our study, suggesting that substantial biological events take place that are far beyond simple proliferation assays (Figure 4). After HGF stimulation, Detroit 562 cells showed regulation of 2850 genes, whereas only 236 and 30 genes were regulated in FaDu and SCC-9, respectively. Importantly, all three cell lines shared only four genes with differences in expression levels after identical HGF stimulation. Compared to copy number variation (CNV) analysis (cbioportal.com), mRNA sequencing and immunoblotting showed significantly higher Met mRNA and protein levels in Detroit 562 (Figure 4b,c).

Enrichment analysis showed activation of signaling pathways and gene set characteristics for HGF stimulation (Table 1 and Table 2). Furthermore, among the significantly upregulated pathways was hsa05230 (Central carbon metabolism in cancer). This gene set assembles genes associated with the Warburg effect. Indeed, Detroit 562 showed strongly altered glycolysis after HGF stimulation (Figure 8a). We could show that two genes encoding key enzymes of glycolysis were significantly upregulated in Detroit 562. In the other two cell lines these effects were smaller (Figure 5a and Figure 6a–d). Although the glycolysis/gluconeogenesis pathway was neither significantly enriched in the set of up- nor in the set of downregulated genes, the changes in these key enzymes seem to be enough for an effect on glycolysis. HK2 has already been shown to be upregulated upon HGF stimulation in a lung cancer model and to promote the proliferation and survival of laryngeal squamous cell carcinoma [11,21]. The LDHA gene encodes the LDHM subunit of lactate dehydrogenase. This enzyme acts at the crossroad between oxidative phosphorylation and glycolysis, as it can catalyze the conversion of pyruvate to lactate. Pyruvate can otherwise be transported to the mitochondrion, where it is converted to acetyl-CoA and is used for preparation of precursors for oxidative phosphorylation in the Krebs cycle. Which reaction a lactate dehydrogenase is catalyzing, pyruvate into lactate or vice versa, depends on its subunit composition. LDH isoforms that contain predominantly subunit M preferably catalyze the conversion of pyruvate into lactate; therefore, a higher level of this isoform can indeed promote glycolysis [22].

Interestingly, HGF stimulation of Detroit 562 resulted in substantial downregulation of alcohol dehydrogenases 1B, 1C, and ADH7 (Figure 5a and Figure 6e). This was also seen in FaDu cells, but to a lesser extent. We could confirm this downregulation for ADH1B on protein level (Figure 6f). In a gene expression study on laryngeal squamous cell carcinoma, a subtype of HNSCC, ADH1C, and ADH7 showed a tumor stage-dependent decreasing expression pattern [23]. Furthermore, in NSCLC, low expression of ADH1B and ADH1C is a prognostic marker for worse overall survival [24]. Even if there are few data available, one can hypothesize that lower ADH1B, ADH1C, and ADH7 levels might influence clinical outcome and overall survival in HNSCC. Further investigation in this field may help us understand the functional relevance of HGF-mediated ADH1B and ADH1C downregulation in HNSCC.

We also observed a link between HGF/Met signaling and FGF/FGFR signaling in our data. HGF signaling strongly influenced the balance between two FGF receptors: FGFR1 expression increased, whereas FGFR3 expression decreased (Figure 5b and Figure 7). The FGFR tyrosine kinase family participates in several aspects of cell biology, including cell survival, angiogenesis, proliferation, migration, and differentiation [25]. Similar to HGF/Met signaling, deregulation of FGFR signaling can lead to cancer progression. In addition, FGFR1 has been shown to have a direct influence on the Warburg effect by phosphorylating glycolytic enzymes pyruvate kinase M2 (PKM2) and pyruvate dehydrogenase kinase 1 (PDHK1) [26,27]. Furthermore, it has been demonstrated that inhibition of FGFR1 signaling in SQCLC cell lines may also have an impact on cancer cell growth by affecting glucose energy metabolism [28]. Further investigations will be necessary to elucidate if FGFR signaling has an impact on glucose metabolism in HNSCC and what the connections to HGF/Met signaling are.

Our findings confirm the results of a study from Kumar and colleagues [29]. They recently reported on HGF-related upregulation of glycolysis in HNSCC cells and further investigated the role of HGF/Met signaling in an in vivo model. Consistent with our findings, they reported an upregulation of HK2, PFK1, and MCT1 in HGF-stimulated HNSCC cell lines. The gene encoding MCT1 was overexpressed in HGF-stimulated Detroit 562 (Appendix A, log_2_ fold increase of 0.44), as well as the gene for PFK1 (the PFKP isoform, log_2_ fold change of 0.44, *p*_adj_ < 0.1, Figure 5a). Furthermore, they found that HGF triggers increased bFGF secretion, which in turn leads to increased oxidative phosphorylation and HGF secretion in cancer-associated fibroblasts (CAFs). Also, bFGF is one of the upregulated genes in Detroit 562, showing a log_2_ fold change of 0.37 (Appendix A, *p*_adj_ < 0.1).

Our data make it possible to directly compare the gene expression of an HNSCC cell line reacting to HGF with glycolysis upregulation to the gene expression of two rather unresponsive ones. Of our tested cell lines only the one with the highest Met receptor level showed a strong metabolic effect. Therefore, the amount of Met receptor expressed by the cells could be crucial. Furthermore, these results suggest that HGF affects not only FGFR-signaling in CAFs, but could also affect FGFR signaling in HNSCC cells themselves. These findings should definitely be explored further by investigating other HNSCC cell lines expressing high levels of Met receptor or tumor samples.

Together with the Kumar study, our study provides evidence that HGF/Met signaling may play an important role in maintaining a central hallmark of cancer, the Warburg effect. We believe that these metabolic changes also result in an immunosuppressive tumor microenvironment. This is relevant for new immunotherapies that have recently entered the field of head and neck cancer treatment.

## 4. Materials and Methods

### 4.1. Cell Lines

HNSCC cell lines were obtained from ATCC. Detroit 562 and FaDu were cultured in MEM-α and SCC-9 in DMEM F12 (+HEPES) containing 400 ng/mL hydrocortisone (Thermo Fisher Scientific, Waltham, MA, United States).

### 4.2. Western Blotting

Cells were seeded in 6-well plates (500,000 cells/mL) and on the next day treated with 50 ng/mL HGF or with 50 ng/mL HGF and 0.5 µM foretinib/PHA-665752 (Selleck Chemicals, Munich, Germany) or were left untreated. In the case of p-MET detection, cells were kept for 30 min at 4 °C after stimulation. For HKII and LDHA detection, cells were incubated at 37 °C for 48 h, for ADH1 detection at 37 °C for 24 h. For sample preparation, cells were washed with 2 mL PBS and scraped off in 200 µL lysis buffer (10 mM Tris-HCl pH 7.6, 5 mM EDTA, 50 mM NaCl, 30 mM Na_4_P_2_O_7_, 50 mM NaF, 1 mM Na_3_V0_3_, 1% Triton x-1000, protease inhibitor mix, and phosphatase inhibitor mix (cOmplete, Roche, Basel, Switzerland). Samples were sonicated with an amplitude of 25% for 20 s (model 4010, Branson, Danbury, United States). The total protein concentration was measured (DC Protein Assay Kit, Bio-Rad, Hercules, CA, United States), and the same amounts of protein per sample were loaded onto 12% SDS gels. After the gel was run and blotted, nitrocellulose membranes were blocked with 5% dry milk in TBS for 1 h. Blots were incubated overnight at 4 °C with primary antibodies anti-p-Met (Cell Signaling, Danvers, United States, #3077, 1:1000 in TBS), anti-total Met (Cell Signaling #3127, 1:100 in TBS with 5% dry milk and 0.1% Tween 20), anti-HKII (Cell Signaling #2867, 1:1000 in TBS with 0.1% Tween 20 and 5% BSA), anti-LDHA (Cell Signaling #3582, 1:1000 in TBS with 0.1% Tween 20 and 5% BSA), anti-ADH1 (Cell Signaling #5295, 1:1000 in TBS with 0.1% Tween 20 and 5% BSA) and anti-tubulin (Thermo Fisher Scientific, Waltham, United States, 1:4000 in TBS). The next day, after washing three times for 5 min with TBS containing 0.1% Tween 20 (TBS-T), blots were incubated with the appropriate HRP-coupled secondary antibodies, washed again 3 times for 10 min with TBS-T, and subjected to signal detection (ECL Western Substrate, Pierce; ChemiDoc Imaging System, Bio-Rad). If applicable, blots were incubated 2 times for 10 min in stripping buffer (0.2 M glycine, 3.5 mM SDS, 1% Tween 20, pH 2.2). After washing (4 times for 5 min with TBS-T), the blots were blocked and reprobed.

### 4.3. Cytotoxicity and Proliferation Assays

For testing the impact of the two tyrosine kinase inhibitors, 10,000 cells per well were seeded in 96-well plates and on the next day were kept untreated or were treated in triplicate with increasing concentrations of foretinib and PHA-665752 (C_max_ = 24 µM and C_max_ = 60 µM, respectively, Selleck Chemicals, Munich, Germany). After 48 h, the supernatant was discarded, and the cells were incubated for 10 min with 50 µL 0.5% (*w*/*v*) crystal violet solution and washed three times using distilled water. Crystal violet was dissolved in 100 µL 98% methanol per well, and absorption at 595 nm was measured (Photometer Infinite F50, Tecan, Männedorf, Switzerland). For determination of IC_50_ values, data were normalized on a scale of 0%–100% viability and fitted with the software Prism (GraphPad, San Diego, United States) using nonlinear regression, log(inhibitor) vs. normalized response (variable slope).

HGF-dependent proliferation was also determined using a crystal violet assay as described above. The cells were seeded at a density of 4000 cells/well in 96-well plates and incubated overnight. The next day, HGF was added, and the cells were incubated for another 72 h until the crystal violet test was performed.

### 4.4. mRNA Sequencing

Cells were seeded in 6-well plates (750,000 cells/well) and on the next day were treated with 50 ng/mL HGF or were left untreated. After 16 h of treatment, the cells were washed with 2 mL cold PBS and detached from the plate using 1 mL of 10% Trypsin-EDTA (Merck, Darmstadt, Germany). Cells were centrifuged for 5 min at 2000 rpm and 4 °C. Total RNA was isolated using a commercially available kit (RNeasy Mini Kit, Qiagen, Hilden, Germany). Cells were lysed using 350 µL RLT lysis buffer, and a syringe and needle were used for homogenization. Further processing was performed as described in the instructions provided by the manufacturer. The experiment was repeated twice.

RNA quality was checked using a 2100 Bioanalyzer with RNA 6000 Nano kits (Agilent, Santa Clara, United States). The RIN for all samples was between 9.6 and 9.9. A total of 0.6 µg of every sample was used to generate libraries with the TruSeq stranded mRNA kit (Illumina, San Diego, United States) according to the manufacturer’s instructions. Libraries were quantified via a Qubit 3.0 Fluorometer (Thermo Fisher Scientific, Waltham, United States), and quality was checked using a 2100 Bioanalyzer with DNA 1000 kits (Agilent). Sequencing of the pooled libraries was performed in single-end mode using High Output Kits v2/2.5 (75 cycles) on the NextSeq 500 platform (Illumina). For each library between 34 and 42 million reads were generated. The sequencing data are available at the Gene Expression Omnibus (GEO) [30] under accession number GSE135552.

To assure high sequence quality, Illumina reads were quality- and adapter-trimmed via Cutadapt [31] version 1.16/1.17 using a cutoff Phred score of 20 in NextSeq mode, and reads without any remaining bases were discarded (command line parameters: --nextseq-trim=20 -m 1 -a AGATCGGAAGAGCACACGTCTGAACTCCAGTCAC).

After trimming, we aligned the resulting reads to the human genome (GRCh38.p12) using STAR [32] version 2.5.2b with default parameters and including GENCODE [33] Human Release 28 transcript annotations to support the mapping.

For all studied samples, the proportion of reads uniquely mapped to the human reference genome ranged between 86% and 88% in total. The sequences aligning to specific gene annotations from GENCODE (see above) were quantified using BEDTools [34] (bedtools intersect -S -wa -c -a GENE_ANNOTATION_FILE -b BAM_FILE). Next, differentially expressed genes between HGF-treated and untreated libraries for each cell line were identified using DESeq2 [35] (version 1.22.1). Here, a count table with read counts for all cell lines was used as input. Read Counts were normalized by DESeq2 and fold-change shrinkage was applied by setting the parameter “betaPrior=TRUE”. For each pairwise comparison, all genes with log_2_ fold change ≥ |+/−0.4| and *p*_adj_ < 0.1 were considered significantly differentially expressed. Heatmaps were generated using the “clustermap” function of the Python library seaborn (version 0.9.0) with linkage method “average” and metric “euclidean” for hierarchical clustering.

Functional enrichment analysis was conducted using the Bioconductor/R package clusterProfiler (3.12.0) [36]. At first KEGG annotations for human pathways (hsa) were downloaded via the clusterProfiler function download_KEGG. Next, Entrez Gene IDs and gene names were mapped to GENCODE gene annotations based on ENSEMBL gene IDs using the mapIds function with the AnnotationDbi (1.46.0) and org.Hs.eg.db (3.8.2) Bioconductor/R packages (Appendix A). Enriched pathways were identified via the clusterProfiler function enricher using Entrez Gene IDs of significantly up- or downregulated genes and downloaded KEGG annotations as input. If more than one Entrez Gene ID was mapped to a GENCODE gene only the first ID in the list was considered. To include all available pathways, also nonsignificant ones, in the output tables, the parameters pvalueCutoff and qvalueCutoff were set to 1.1.

### 4.5. qPCR

Stimulation and RNA isolation was performed as described for mRNA sequencing. First, 1 µg RNA was subjected to cDNA generation using the QuantiTect Reverse Transcription Kit (Qiagen, Hilden, Germany) according to the instructions provided by the manufacturer. Next, 20 ng cDNA was used in the PCR reaction with 1.5 µL of the appropriate primer (QuantiTect Primer Assay, Qiagen) and 12.5 µL of a ready-to-use qPCR master mix (QuantiTect SYBR Green PCR Kit, Qiagen). The thermal cycling program was composed of initial denaturation at 95 °C for 15 min, 39 cycles at 95 °C for 15 s, 30 s at 54 °C, and 30 s at 72 °C. Duplicates for each data point were measured. The gene for β-actin was used as internal control and the 2^−ΔΔCt^ method [37] for relative quantification of gene expression.

### 4.6. Measuring Glycolysis with an Extracellular Flux Analyser

To measure differences in glycolysis, glycolysis rate assays were performed using a Seahorse XF 96 analyzer (Agilent, Santa Clara, CA, United States). Assay medium was DMEM without phenol red (D5030, Merck, Darmstadt, Germany), supplemented with 5 mM HEPES, 10 mM glucose, 2 mM l-glutamine, and 1 mM pyruvate. Using 1 N NaOH, the pH was adjusted to 7.4. A total of 15,000 cells/well were seeded in 96-well XF cell culture plates using standard medium and stimulated with 50 ng/mL HGF alone or in combination with 50 µg/mL PHA the next day. Cells treated only with medium were used as controls. For each tested condition, 10 wells were seeded. After 48 h, the assay was performed as described by the manufacturer. The medium on the cells was removed, and the cells were washed with 200 µL assay medium twice and incubated with 200 µL assay medium at 37 °C for 1.5 h. Directly before the start of the assay, assay medium was removed, and 180 µL fresh, warm assay medium was added. The following final concentrations of inhibitors were used: 0.5 µM rotenone/antimycin A (Merck, Darmstadt, Germany) and 50 mM 2-deoxy-d-glucose (2-DG, Merck, Darmstadt, Germany). After the measurement, cells were fixed using 75% ethanol and stained with 0.1% (*w*/*v*) crystal violet in 20% ethanol. The assay plate was washed with ddH_2_O and dried at room temperature. Absorption was measured at 595 nm using 10% acetic acid, and results were used for normalization of the data.

## Figures and Tables

**Figure 1 ijms-21-00471-f001:**
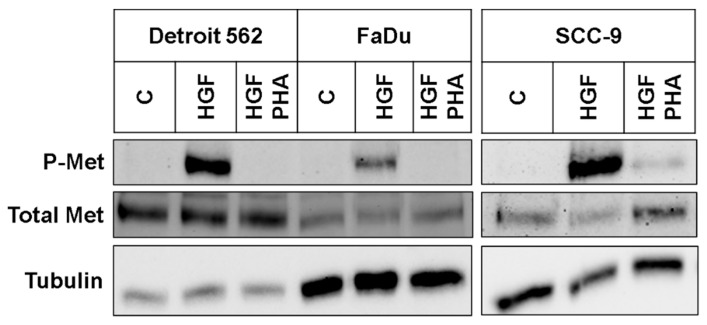
Hepatocyte growth factor (HGF) activates the Met receptor on the investigated cell lines. Results of immunoblotting using an antibody recognizing Met receptors phosphorylated at Tyr1234/Tyr1235 (P-Met). Cells were treated for 30 min with 50 ng/mL HGF (HGF), HGF plus 50 µM PHA-665752 (HGF PHA), or neither (C). Blots were stripped and reprobed using a Met-specific antibody recognizing total Met. As a loading control, lower parts of the blots were incubated with an antibody recognizing tubulin. SCC-9 samples were loaded onto a separate gel. A typical experiment out of three is shown.

**Figure 2 ijms-21-00471-f002:**
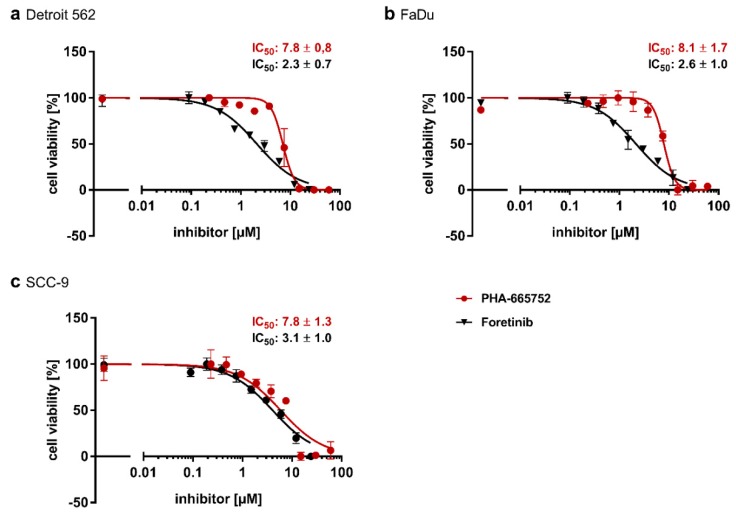
Concentration-dependent effect of two Met kinase inhibitors on viability of cell lines Detroit 562 (**a**), FaDu (**b**), and SCC-9 (**c**). Cells were incubated with different concentrations of tyrosine kinase inhibitors PHA-665752 and foretinib or treated with medium alone (first datapoint on the left) for 48 h. Afterwards, cells were stained with crystal violet, and absorption at 595 nm was measured. Data were fitted with GraphPad Prism using nonlinear regression, log(inhibitor) vs. normalized response (variable slope), resulting in the IC_50_ values shown next to the graphs. IC_50_ values are means with SD of at least three independent experiments.

**Figure 3 ijms-21-00471-f003:**
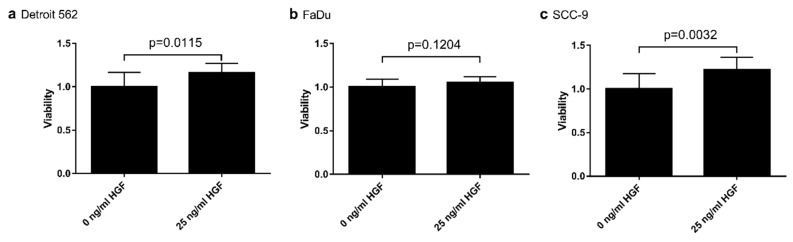
Effect of HGF on proliferation of cell lines Detroit 562 (**a**), FaDu (**b**), and SCC-9 (**c**). Cells were treated with or without HGF for 72 h. Afterwards, cells were stained with crystal violet, and absorption at 595 nm was measured. Result of untreated cells was set as one.

**Figure 4 ijms-21-00471-f004:**
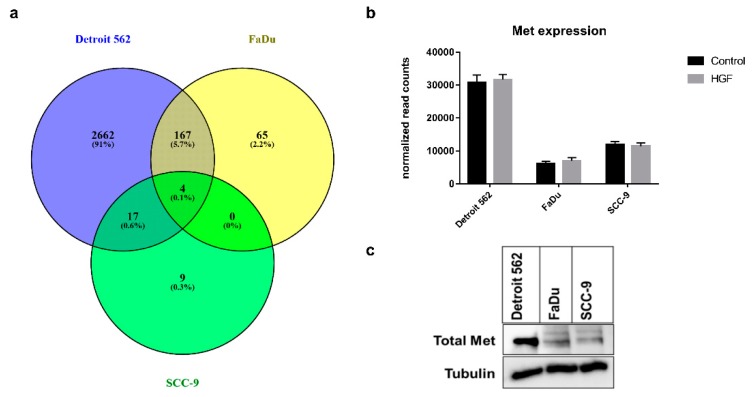
(**a**) Venn diagram showing the differences and overlap in induced or downregulated genes between the three cell lines upon 16 h of HGF stimulation (50 ng/mL). Genes with a difference in expression of ≥ 0.4 or ≤ -0.4 (log_2_ fold change) and *p*_adj_ < 0.1 (see Materials and Methods) were considered induced/downregulated. (**b**) Diagram showing mRNA sequencing-based normalized gene expression levels for the gene MET. HGF-stimulated samples in grey, unstimulated control in black. (**c**) Immunoblotting showing the protein expression of MET in the investigated cell lines (Total Met). As a loading control, the blot was reprobed using an antibody recognizing tubulin. A typical experiment out of three is shown.

**Figure 5 ijms-21-00471-f005:**
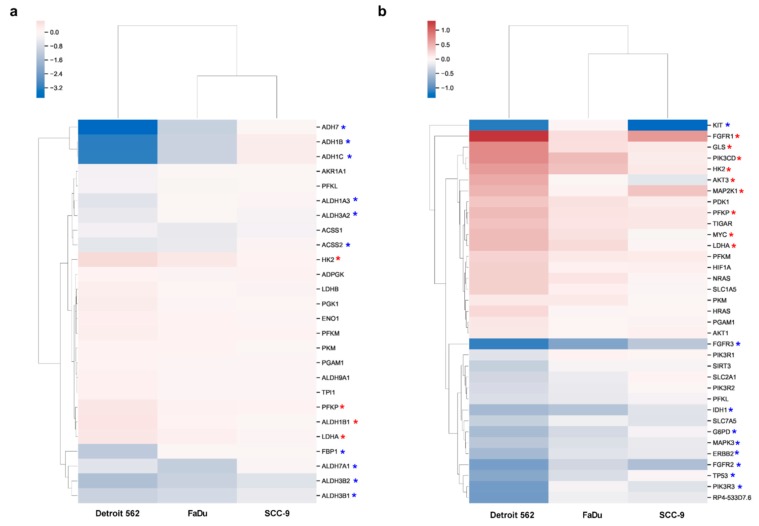
Heat maps of mRNA-sequencing results showing genes participating in KEGG pathway hsa00010, glycolysis/gluconeogenesis (**a**) or KEGG pathway hsa05230, central carbon metabolism in cancer (**b**). The color scale indicates log_2_ fold changes between the HGF-stimulated sample and the unstimulated control. Upregulated genes are depicted in red, downregulated genes in blue colors, and unaltered genes in white. Only pathway-specific genes with *p*_adj_ < 0.1 in Detroit 562 are shown. Red asterisks indicate genes with a log_2_ fold change ≥ 0.4, blue asterisks indicate genes with a log_2_ fold change ≤ −0.4.

**Figure 6 ijms-21-00471-f006:**
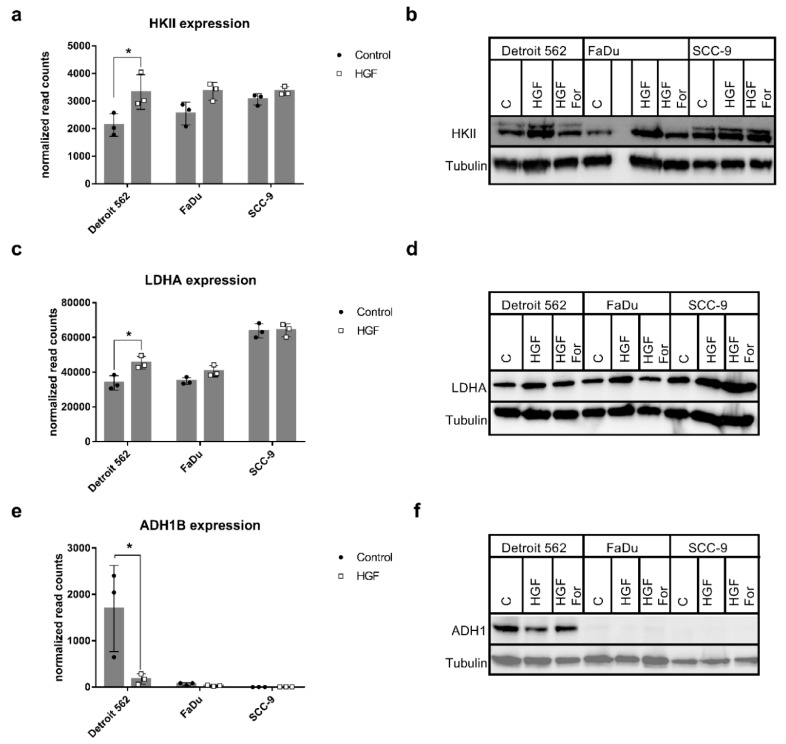
Differential expression of genes participating in glycolysis/gluconeogenesis. (**a**,**c**,**e**) Diagrams show normalized gene expression levels resulting from mRNA sequencing for genes HKII (**a**), LDHA (**b**), and ADH1B (**e**). HGF-stimulated samples are shown in grey, untreated controls in black. *: *p*_adj_ < 0.1. *(***b**,**d**,**f**) Western blots with lysates of cells stimulated for 48 h (24 h in case of ADH1) with 0.6 nM HGF (HGF), HGF plus 500 nM foretinib (HGF For), or neither (C) for the indicated proteins. Blots were stripped and reprobed with an antibody specific for tubulin to show differences in gel loading. A typical experiment out of three is shown.

**Figure 7 ijms-21-00471-f007:**
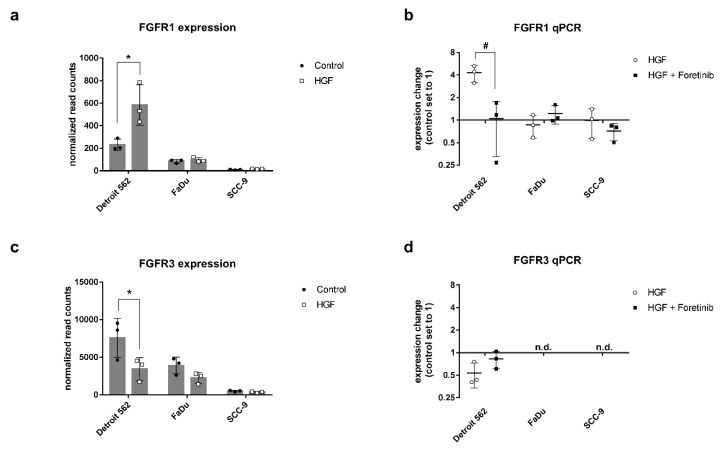
Differential expression of genes of the FGFR family. (**a**,**c**) Diagrams show normalized gene expression levels resulting from RNA sequencing for FGFR1 (**a**), and FGFR3 (**c**). HGF-stimulated samples are shown in grey, untreated controls in black. *: *p*_adj_ < 0.1. (**b**,**d**) qPCR results for FGFR1 and FGFR3. RNA was isolated after 20 h of stimulation with 0.6 nM HGF or HGF and foretinib (500 nM). Shown are expression changes in relation to the untreated control, which was set to one. #: *p* < 0.05. n.d.: not detectable.

**Figure 8 ijms-21-00471-f008:**
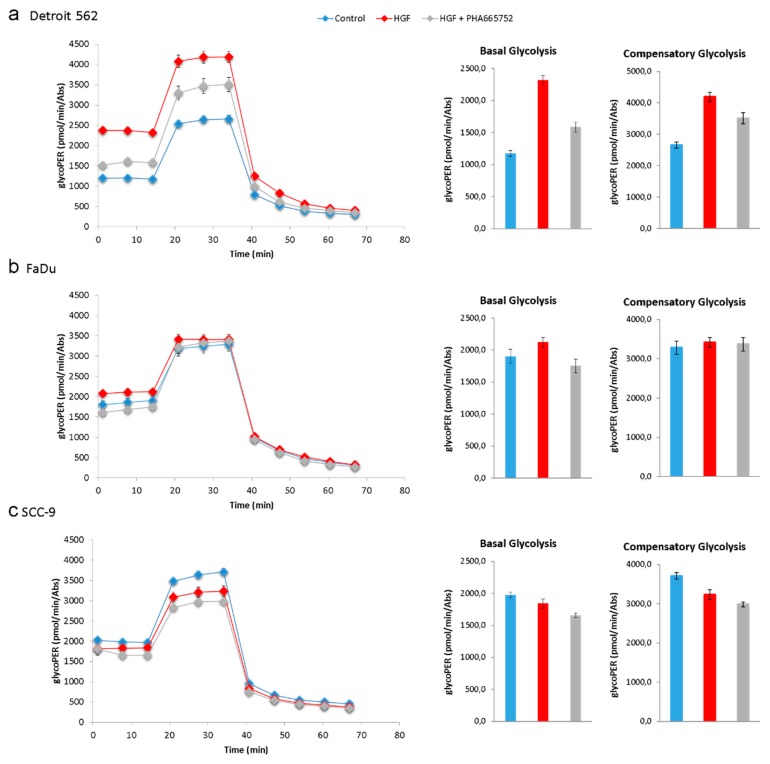
Glycolytic proton efflux is increased upon HGF stimulation in Detroit 562 cells. Detroit 562 (**a**), FaDu (**b**) and SCC-9 cells (**c**) were stimulated with 50 ng/mL HGF (red), 50 ng/mL HGF, and 50 µg/mL PHA-665752 (grey) or neither (Control, blue). After 48 h, the glycolytic proton efflux rate (glycoPER) was measured. On the left side, kinetic graphs are shown with eleven datapoints from left to right. First, basal glycolysis was measured (first three datapoints), and mitochondrial function was inhibited using an injection of rotenone A and antimycin A (after measurement 3). Measurements 4–6 were used to measure compensatory glycolysis. To prove that the measured proton efflux was mainly produced by glycolysis, glycolysis inhibitor 2-DG was injected after measurement 6, resulting in a rapid decline of proton efflux (datapoints 7–11). On the right side, glycoPER values for basal and compensatory glycolysis are shown as bar charts (corresponding to values of datapoints 3 and 6 in the kinetic graphs). Raw data were normalized using absorbance at 595 nm of crystal violet-stained cells. Datapoints are means with SEM, *n* = 10. A typical experiment out of at least three is shown.

**Table 1 ijms-21-00471-t001:** Upregulated signaling pathways upon HGF stimulation in Detroit 562. Ranking based on adjusted *p*-value (*p*.adjust). Results of the first seven pathways on the list and of two pathways of interest are shown. Only genes that are part of at least one of the pathways listed in KEGG and that show a difference in expression of ≥ 0.4 (log_2_ fold change) and *p*_adj_ < 0.1 are included.

Rank	ID ^1^	Description	GeneRatio ^2^	BgRatio ^3^	*p*.adjust ^4^
1	hsa03008	Ribosome biogenesis in eukaryotes	22/408	105/7946	3.3 × 10^−6^
2	hsa05205	Proteoglycans in cancer	30/408	204/7946	2.1 × 10^−5^
3	hsa04510	Focal adhesion	26/408	199/7946	0.0009
4	hsa04010	MAPK signaling pathway	33/408	295/7946	0.0013
5	hsa04012	ErbB signaling pathway	14/408	85/7946	0.0054
6	hsa04151	PI3K-Akt signaling pathway	35/408	354/7946	0.0063
7	hsa05222	Small cell lung cancer	14/408	92/7946	0.0084
**33**	**hsa05230**	**Central carbon metabolism in cancer**	**9/408**	**69/7946**	**0.0707**
**186**	**hsa00010**	**Glycolysis/Gluconeogenesis**	**4/408**	**68/7946**	**0.7171**
**Total 278**					

^1^ ID: identification number of the pathway at Kyoto Encyclopedia of Genes and Genomes (KEGG). ^2^ GeneRatio: number of upregulated genes in this pathway/total number of upregulated genes. ^3^ BgRatio: total number of genes in this pathway/total number of genes in KEGG. ^4^
*p*.adjust: adjusted *p* value for indicated pathway.

**Table 2 ijms-21-00471-t002:** Downregulated signaling pathways upon HGF stimulation in Detroit 562. Ranking based on adjusted *p*-value (*p*.adjust). Results of the first seven pathways on the list and of two pathways of interest are shown. Only genes that are part of at least one of the pathways listed in KEGG and that show a difference in expression of ≤ −0.4 (log_2_ fold change) and *p*_adj_ < 0.1 are included.

Rank	ID ^1^	Description	GeneRatio ^2^	BgRatio ^3^	*p*.adjust ^4^
1	hsa05204	Chemical carcinogenesis	28/763	82/7946	2.7 × 10^−7^
2	hsa00980	Metabolism of xenobiotics by cytochrome P450	24/763	76/7946	1.2 × 10^−5^
3	hsa00053	Ascorbate and aldarate metabolism	12/763	27/7946	0.0002
4	hsa00860	Porphyrin and chlorophyll metabolism	15/763	42/7946	0.0003
5	hsa00140	Steroid hormone biosynthesis	18/763	60/7946	0.0005
6	hsa05322	Systemic lupus erythematosus	29/763	133/7946	0.0009
7	hsa00982	Drug metabolism—cytochrome P450	19/763	72/7946	0.0014
**86**	**hsa00010**	**Glycolysis/Gluconeogenesis**	**10/763**	**68/7946**	**0.4096**
**113**	**hsa05230**	**Central carbon metabolism in cancer**	**9/763**	**69/7946**	**0.5790**
**Total 311**					

^1^ ID: identification number of the pathway at Kyoto Encyclopedia of Genes and Genomes (KEGG). ^2^ GeneRatio: number of upregulated genes in this pathway/total number of upregulated genes. ^3^ BgRatio: total number of genes in this pathway/total number of genes in KEGG. ^4^
*p*.adjust: adjusted *p* value for indicated pathway.

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
