# Peer review of "The Influence of Met Receptor Level on HGF-Induced Glycolytic Reprogramming in Head and Neck Squamous Cell Carcinoma"

_ijms, 2020, doi:10.3390/ijms21020471_

Round 1

Reviewer 1 Report

Thank you for your detailed and professional responses to my comments. I found that my concerns have been adequately addressed by the authors, and I would like to recommend this manuscript for publication on IJMS.

Just a note for the proof: Figure 5 seems to be incorrectly rotated in the review version.

Author Response

Dear Reviewer,

thank you very much for recommending or manuscript for publication.

In our newest revised version we had to correct the venn diagram in figure 4 as there has been an error during preparation. It now shows the correct data and up- as well as downregulated genes were included. The sections in the text dealing with this diagram were changed accordingly.

Furthermore, due to the recommendation of another reviewer we performed a more global analysis of our mRNA sequencing data, a functional enrichment analysis. We added two tables showing a list of the pathways which are upregulated (table 1) and downregulated (table 2) the most in Detroit 562 and the results of the two pathways mentioned in the manuscript. Full data for this analysis can be found in new supplementary data S4-S9.

We rotated figure 5, as we wanted to make it easier to read and now changed the corresponding page to horizontal. We also added asterisks to indicate the genes with the highest expression changes.

Thank you very much for your very helpful reviews.

Best regards,

the authors

Reviewer 2 Report

It was not easy to follow the Authors' revised version without the "response to reviewer" document. Authors just highlighted some parts that is not clear they edited the text or added.

Authors hypothesized that activation of HGF/Met signaling in HNSCC influences glucose metabolism. As I mentioned before, I am unfortunately not seeing the current results and conclusions as coming together to provide sufficient detailed biological and mechanistic insights into the role of HGF/Met signaling in metabolic reprogramming of HNSCC. In addition, the current data and analyses do not seem sufficiently comprehensive to support the main conclusions. Regarding to my previous comments, Authors just added one sentence in discussion section “These findings should definitely be explored further by investigating other HNSCC cell lines expressing high levels of Met-receptors or tumour samples” that is not convincing.

The gene expression analysis is also so naïve. I highly recommend to use an enrichment analysis tool e.g., GSEA (http://software.broadinstitute.org/gsea/index.jsp) to determine whether an a priori defined set of genes shows statistically significant.

Author Response

Dear Reviewer,

thank you very much for your suggestions on our manuscript. Due to the extensive revision with time consuming experiments we couldn´t submit a “response to reviewer document” to our last manuscript version. Our manuscript had to be handled as a new submission. The answers to the reviewers were added to the “letter to the editor”, as we were instructed to do so. It looks like you didn´t received it, though. We therefore repeat some of our points from our last answer below.

As we already pointed out in our reply to your latest review, trying to cover a range of conditions, we examined three cell lines with different levels of Met receptor. Unfortunately, although overexpression of MET receptor in tumor samples is quite common (Seiwert et al., 2009), commercially available cell lines showing amplification of the gene for the Met receptor are rare to find. In a genetic study including a set of 31 tumor cell lines of the upper respirational tract, only cell line Detroit 562 showed this amplification (Barretina et al., 2012). This cell line is already part of our investigation. Chance is high, that if we add other cell lines randomly, the manuscript won´t benefit, as we would probably not have added a high expressing one. Of course we would be able to check a lot of cell lines for their MET expression, but we think this would go beyond the scope of this revision. We therefor apologize for not being able to perform these additional experiments. Of course, you are right, three cell lines are quite a low number to make the claim that glycolytic reprogramming depends on Met receptor level. We therefor changed our title and state in our discussion that investigation of additional cell lines is needed. We agree that a global analysis of our mRNA sequencing data is missing in our manuscript, so we performed an enrichment analysis. We added two tables showing a list of the pathways which are significantly upregulated (table 1) and downregulated (table 2) the most in Detroit 562 and the results of the two pathways mentioned in the manuscript. Full data for this analysis can be found in new supplementary data S4-S9. In our newest revised version we had to correct the venn diagram in figure 4 as there has been an error during preparation . It now shows the correct data, and up- as well as downregulated genes were included. The sections in the text dealing with this diagram were changed accordingly.

We want to point out that for our last revision we provided additional biochemical data, confirming the mRNA sequencing data (Fig. 6 and 7). Furthermore, the enrichment analysis showed, that pathway hsa05230 (Central carbon metabolism in cancer) is significantly enriched in Detroit 562, so we think it is definitely feasible to say that our data show that HGF plays a role in metabolic reprogramming.

Best regards,

the authors

Round 2

Reviewer 2 Report

Thanks for performing the enrichment analysis that improved and validated the results. The Authors addressed all my concerns and manuscript is ready to be published.

This manuscript is a resubmission of an earlier submission. The following is a list of the peer review reports and author responses from that submission.

Round 1

Reviewer 1 Report

I appreciate that this manuscript explores the Warburg effect and HGF/Met signalling in the HNSCC. However, based on my experience with broadly related papers, I am unfortunately not seeing the current results and conclusions as coming together to provide sufficient detailed biological and mechanistic insights into the role of HGF/Met signalling in metabolic reprogramming of HNSCC. Also, the current data and analyses do not seem sufficiently comprehensive to compellingly support the main conclusions.

I suggest that Authors should investigate this hypothesis at least on two other cell lines.

Reviewer 2 Report

Verena Boschert et al in this study described the effects of HGF in three HNSCC cell lines on glucose metabolism and proliferation, they concluded that HGF-induced upregulation of three key glycolytic genes and the resulting increased glycolysis was associated with the Met expression levels. The conclusion is not solid or convincing. The manuscript is overall interesting, however, it lacks the novelty.

some comments:

Since this study is focused on the HNSCC, the introduction should be more clear on the HNSCC,  Met overexpression in 90% HNSCC or human tissue samples?   Ligand HGF was found to be overexpressed in almost 50 % of human tissue, so what about the HNSCC?  Did HNSCC  or the surrounding tissues/ cells express HGF?  is there any mutation associated with the abnormal expression of Met in HNSCC?  In three HNSCC cells, HGF/Met signaling mediated increased glycolysis, presumably mediated by higher expression of three key enzymes of glycolysis and associated with the MEt expression levels. To validate this conclusion, another WB should be done to show the total Met and P-Met in the same gel and in the equal total protein load in Fig.1, instead of just shown equal load of tubulin in Fig.4C. The authors should explain in the three cell lines, why SCC-9 with low total Met, no changes of glycolytic genes upon HGF, but with a strong effect on proliferation; FaDu with low total Met, no changes of glycolytic genes upon HGF and no effect on proliferation? Besides the RNA changes shown by RNA-seq, the WB needs to show the protein levels in Fig. 6.  Fig.7 is not easy to understand when across the three cell lines to support their conclusion on carbon metabolism.

Reviewer 3 Report

This interesting manuscript shows how induction with HGF can cause changes in glycolysis in HNSCC, arguably via a Met-related pathway. I found that the current evidences are good, but not sufficient to make such broad claim. I would like to see the manuscripts revised to be more precise, and additional data added. My point-by-point comments are as below:

- While Met was shown to be activated by HGF treatment, and HGF treatment altered multiple pathways, there is no evidence in the manuscript showing that those effects depend on Met. If this conclusion is to be drawn, please do experiment in which effects of HGF treatment are assessed when Met expression/activity is modulated.

- In figure 2, please provide information on how the IC50s were calculated (i.e. GraphPad parameters) and check to make sure that the IC50 for Foretinib in SCC-9 is actually 1.8. Based on the graph, the IC50 should be much higher than that.

- In figure 6 and 7, please confirm the RNA-seq data with qPCR and western blotting. Also, for figure 7, the mRNA expression of FGFR does not matter if the phosphorylation does not change. Please provide Western blot data to confirm the change in activation of this receptor.

- For figures with bar-graph, please show individual data points (Figures 3, 4b, 6a, 6c-f, 7, 8). https://www.graphpad.com/support/faq/graph-tip-how-can-i-make-a-barcolumn-graph-that-also-shows-the-individual-data-points/

- Please provide higher quality versions for Figure 5a-b.

- Please provide citations for lines 57-61.

- Please consider breaking up the big paragraph lines 112-147 to make the manuscript easier to read.

- A suggestion: Using the full phrase instead of the HNSCC abbreviation in the title may make it easier for readers to find your paper.